# Missed Insights for Earlier Management of Parkinson’s Disease and the Value of Dopamine Transporter (DAT) Scans

**DOI:** 10.3390/geriatrics9050126

**Published:** 2024-10-01

**Authors:** Mohib Hafeez, Elizabeth Eoff, Jeanne Wei, Gohar Azhar

**Affiliations:** 1Donald W. Reynolds Department of Geriatrics, Institute on Aging, University of Arkansas for Medical Sciences, Little Rock, AR 72205, USA; eleoff@uams.edu (E.E.); weijeanne@uams.edu (J.W.); azhargohar@uams.edu (G.A.); 2College of Medicine, University of Arkansas for Medical Sciences, Little Rock, AR 72205, USA

**Keywords:** Parkinson’s disease, dopamine transporter scan, cognitive impairment, falls, early diagnosis, neuroimaging, geriatric neurology

## Abstract

**Background/Objectives:** This retrospective study focused on the role of Dopamine Transporter (DAT) scans in diagnosing Parkinson’s Disease (PD) in older adults with cognitive impairment (CI). **Methods:** We retrospectively analyzed brain imaging of 6483 individuals aged 60 and above with CI. Among these, 297 underwent a DAT scan, with 189 testing positive and 89 starting dopamine therapy. In contrast, 173 patients exhibited PD-associated structural changes on CT or MRI without receiving DAT scans or treatment. **Results:** Of these patients, 50 (29%) experienced falls. This points towards a potential missed diagnosis of PD, which can respond to therapy in the early stages. **Conclusions:** Our results suggest that providers may overlook subtle signs of parkinsonism in patients with CI, resulting in symptoms worsening and treatment delay. Since CI is often first brought to the attention of PCPs, our findings call for an increased effort to inform PCPs of the role of DAT scans in aiding the diagnosis of dopamine deficiency states. By understanding PD-related structural changes seen on brain imaging and using a DAT scan to confirm dopamine deficiency, treatment for PD or related states might be started earlier or a timely referral made to a specialist, reducing patient disability and improving their quality of life.

## 1. Introduction

Parkinson’s Disease (PD) is a progressive neurodegenerative disorder characterized by motor symptoms such as masked facies, bradykinesia, rigidity, tremors, slowness of gait, and balance problems, as well as non-motor symptoms including fatigue, cognitive impairment, and psychosis [1]. Many of these signs and symptoms of Parkinson’s in the early stages are often attributed by the patients as well as their primary care physicians to the simple slowing down caused by aging, especially in patients without a significant tremor. Patients with early-stage Parkinson’s and memory impairment are often misdiagnosed as having Alzheimer’s disease or Alzheimer’s-disease-related dementia (ADRD). This challenge is further compounded by the difficulty in differentiating PD from atypical parkinsonisms—such as multiple system atrophy (MSA) and progressive supranuclear palsy (PSP)—which share overlapping symptoms in the early stages [2]. Dysautonomia, or dysfunction of the autonomic nervous system, is a critical factor that can aid in distinguishing PD from atypical parkinsonisms. Autonomic dysfunctions, such as orthostatic hypotension, gastrointestinal disturbances, and urinary symptoms, are more severe and start earlier in atypical parkinsonisms compared to PD [2,3]. This aspect is particularly relevant in clinical practice, as it provides important clues for differential diagnosis, potentially reducing the rate of misdiagnosis. Affecting over 1% of the population over 60 years of age, PD poses significant challenges for early and accurate diagnosis, which is crucial for effective disease management and improved patient outcomes [4].

Dopamine Transporter (DAT) imaging has emerged as a highly sensitive and specific tool for diagnosing PD and differentiating it from other Parkinsonian disorders. According to Kägi et al. [5], “DAT-SPECT can prove or exclude with high sensitivity nigrostriatal dysfunction in those cases and facilitates early and accurate diagnosis”.

Despite these advancements, many patients remain underdiagnosed, particularly those with cognitive impairment, leading to missed diagnoses and delayed treatment, exacerbating adverse outcomes such as gait abnormalities and falls [6,7,8].

This study aims to address this gap by evaluating the role and efficacy of DAT scans in diagnosing PD among cognitively impaired elderly patients. We emphasize the need for heightened awareness and utilization of these diagnostic tools in clinical practice. By demonstrating the diagnostic accuracy and prognostic value of DAT imaging, this study highlights the necessity of DAT scans for early detection, which, when combined with timely and effective therapeutic intervention, can minimize adverse outcomes such as falls in this vulnerable population. Additionally, we seek to compare the incidence of falls between patients who received DAT scans and those who did not, hypothesizing that untreated dopamine deficits in cognitively impaired elderly patients lead to a higher incidence of falls.

Beyond its diagnostic capabilities, DAT imaging has been shown to have prognostic value in PD. Studies demonstrated that lower striatal DAT binding at baseline is associated with a higher risk of reaching clinical milestones, which include motor-related disability, cognitive impairment, and psychosis. Ravina et al. [9] highlight that “lower striatal binding at baseline was independently associated with higher risk for clinical milestones and measures of disease severity, including motor-related disability, falling and postural instability, cognitive impairment, psychosis, and clinically important depressive symptoms”. Brooks [10] supports this view, noting that PET and SPECT can follow the loss of dopamine terminal function in PD, providing a tool for monitoring disease progression objectively.

Dopamine deficiency syndromes encompass a range of conditions characterized by reduced dopamine levels in the brain including PD, PSP, MSA, and Corticobasal Degeneration (CBD). Notably, in patients with cognitive impairment, accurate diagnosis of these syndromes is crucial for effective management and treatment [11]. While DAT scans are highly sensitive and specific for detecting nigrostriatal cell loss, which is a hallmark of PD [12], it is important to recognize that CBD, when secondary to an AD pathology, may present normal DAT scans due to preserved striatal presynaptic Dopamine Transporters, especially early in the disease course [13,14]. A high degree of clinical suspicion in the interpretation of DAT scans is warranted in this context. Even though DAT scans lack high specificity for other dopamine deficiency syndromes, they are highly sensitive to detecting dopaminergic deficits indicating a potential underlying neurodegenerative cause, such as PD. This sensitivity aids in distinguishing and diagnosing different dopamine deficiency syndromes.

DAT scans are also effective in differentiating dementia with Lewy bodies (DLB) from other forms of dementia, such as Alzheimer’s disease. In DLB, there is a significant reduction in striatal DAT binding, similar to in PD, which is not observed in Alzheimer’s disease. This distinction is pivotal as it influences treatment strategies and prognosis [15].

DAT scans play a vital role in the timely identification of dopamine deficiency syndromes and help narrow down differentials in complex cases. Their high sensitivity in detecting dopaminergic deficits allows for early diagnosis and intervention, improving patient outcomes and quality of life. As noted by Williams and Lees [11], “Moreover, for patients for whom the diagnosis is unclear, clinicians must continue to describe accurately the clinical picture of each individual, rather than label them with inaccurate diagnostic categories, such as atypical parkinsonism or PSP mimics. In this way, the development of the clinical features can be informative in assigning less common nosological categories that give clues to the underlying pathology and an understanding of the expected clinical course”. This is particularly important in the context of PSP-Parkinsonism Predominant (PSP-P), which can be challenging to differentiate from PD in its early stages. PSP-P shares several clinical features with PD, including Parkinsonian symptoms and a response to dopaminergic therapy, making early diagnosis difficult. Recent advancements in neuroimaging techniques, such as Magnetic Resonance Imaging (MRI) and Single-Photon Emission Computed Tomography (SPECT), using HMPAO, have been shown to assist in distinguishing PSP-P from PD by highlighting differences in brain perfusion, particularly within the frontal lobe and thalamus [16]. In this context, DAT scans fill the role of first-line imaging in the identification of dopaminergic deficits, with subsequent specialized imaging used to obtain a personalized diagnosis.

### Practical Applications

In clinical practice, the incorporation of DAT imaging has significantly improved the accuracy of PD diagnoses and facilitated better patient management. It is particularly beneficial for patients who do not fully meet the diagnostic criteria of PD or present atypical findings [17]. Ba and Martin [18] emphasized that “DAT-SPECT can accurately detect presynaptic dopaminergic deficits, making it useful in the early diagnosis of PD and differentiation from non-degenerative parkinsonian disorders such as essential tremor, dystonic tremor, and drug-induced parkinsonism”.

The primary objective of this study was to evaluate the role and efficacy of DAT scans in diagnosing PD among cognitively impaired elderly patients, with a focus on the potential impact of early diagnosis and treatment on adverse outcomes such as falls.

## 2. Materials and Methods

### 2.1. Study Design

We retrospectively analyzed a de-identified patient dataset to evaluate the role and efficacy of Dopamine Transporter (DAT) scans in diagnosing Parkinson’s Disease (PD) among cognitively impaired elderly patients. To ensure the accuracy of the DAT scan results, all patients were required to undergo a mandatory washout period of at least two weeks for specific medications known to impact DAT binding. These medications included the following:Stimulants and Related Drugs:–Cocaine;–Amphetamine and methamphetamine;–Methylphenidate;–Modafinil;–Ephedrine;–Phentermine.Antidepressants:–SSRIs and SNRIs (e.g., fluoxetine, sertraline, citalopram, paroxetine);–Bupropion (Wellbutrin, Zyban).Antipsychotics–Haloperidol;–Risperidone.Dopamine agonists–Carbidopa;–Levidopa.Other Medications:–Benztropine;–Mazindol;–Buspirone;–Amoxapine;–Norephedrine;–Phenylpropanolamine.

These drugs can influence DAT binding and potentially confound the results of the scan. UAMS adheres to a strict protocol that mandates this washout period; if the patient did not adhere to this protocol, the DAT scan was not performed.

### 2.2. Study Subjects

This was a retrospective analysis of de-identified patients with CI that was approved by Institutional Board Review, IRB # 262052. The study population consisted of 6483 cognitively impaired patients aged 60 and above who had variable symptomatology of Parkinson’s Disease noted on the problem list. Data were also collected on falls and brain imaging, including CT/MRI or DAT scans. All data were collected over a two-year period, January 2018–December 2021. To ensure the accuracy of the cognitive impairment classification and reduce confounding variables, patients with primary or secondary neoplasms and those with a history of significant trauma were excluded from the study. This exclusion criterion helped us to focus on patients with cognitive impairment due to neurodegenerative or vascular causes. Patients were categorized into two groups:**Group 1 (DAT+ group):** Patients who underwent at least one DAT scan and were diagnosed with Dopamine Transporter deficits.**Group 2 (Missed Diagnosis group):** Patients who displayed PD-associated changes in brain structures (via CT or MRI imaging) but did not receive a DAT scan, PD diagnosis, or dopamine therapy.

### 2.3. Data Analysis

Statistical methods were used to evaluate the effectiveness of DAT scans in diagnosing PD and initiating appropriate treatment. The analysis focused on the following areas:**Diagnostic Accuracy:** The sensitivity and specificity of DAT imaging in identifying PD-related dopaminergic deficits.**Comparison of Outcomes:** A comparison of incidence of falls and other adverse outcomes between the DAT+ group and the Missed Diagnosis group.**Relative Risk (RR):** Calculation of the relative risk of falls in the DAT+ group compared to the Missed Diagnosis group and vice versa.–**RR of Falls in the DAT+ group vs. Missed Diagnosis group:** RR = 0.11 (95% confidence interval (CI): 0.047 to 0.256).–**RR of Falls in the Missed Diagnosis group vs. DAT+ group:** RR = 9.10 (95% CI: 3.90 to 21.23).

Data filtering, counting, and qualitative data analysis were conducted using MS Office, Pycharm, and SPSS (Statistical Package for the Social Sciences) software. Descriptive statistics were calculated and comparison analysis was performed to summarize the data and identify significant differences between the groups. Model Wald chi-squares were used to compare categorical variables, and confidence intervals were calculated to determine the significance of the relative risks.

### 2.4. Ethical Considerations

This study was conducted using de-identified patient data, ensuring patient confidentiality and compliance with ethical standards. All procedures adhered to the principles outlined in the Declaration of Helsinki.

## 3. Results

### 3.1. Role of DAT Scans in PD Diagnosis

Out of the total 6483 cognitively impaired older patients included in the study, 297 patients underwent at least one DAT scan.

**Positive DAT Results:** 189 patients (64%) showed positive results indicative of PD.**PD Diagnosis:** All 189 patients with positive DAT scans were subsequently diagnosed with PD.

These findings highlight the significant role of DAT scans in diagnosing PD, particularly among patients with cognitive impairment where clinical diagnosis alone may be challenging. As Kägi et al. [5] stated, “DAT-SPECT imaging supports the diagnosis of PD or other neurodegenerative parkinsonism in early disease or uncertain or incomplete parkinsonian syndromes”.

### 3.2. Use of Dopamine Therapy

Population subsets of the 189 cognitively impaired patients with positive DAT scans are shown in (Figure 1).

**Dopamine Therapy Initiation:** 89 patients (47%) received Carbidopa–Levodopa, Rasagiline, Ropinirole, Selegiline, Rotigotine, Amantadine, Nourianz, or Pramipexole therapy following their positive DAT scan results.

These data highlight the impact of DAT imaging on clinical decision-making and the initiation of appropriate treatment for PD. Ravina et al. [9] emphasize that “DAT imaging shortly after the diagnosis of PD is associated with clinically important long-term motor and nonmotor outcomes”.

### 3.3. Potential Underdiagnosis of PD

In the Missed Diagnosis group (Figure 1), the following applied:**PD-Associated Brain Changes:** 173 patients exhibited PD-associated changes in brain structures on CT or MRI imaging but did not receive a DAT scan, PD diagnosis, or dopamine therapy.**Falls:** 50 patients (29%) in the Missed Diagnosis group experienced falls compared to 6 patients (3%) in the DAT+ group.

These results suggest that many patients with potential PD were underdiagnosed and did not receive appropriate treatment, leading to a higher incidence of falls and associated adverse outcomes. Palermo and Ceravolo [19] note that “DAT imaging can reveal nigrostriatal impairment even in isolated/atypical tremors, predicting clinical conversion to fully blown parkinsonism”.

### 3.4. Statistical Analysis

The statistical analysis reveals that the DAT+ group had a significantly lower risk of falls compared to the Missed Diagnosis group, indicating the importance of early diagnosis and treatment of PD in reducing adverse outcomes.


**Relative Risk (RR) of Falls:**
–**DAT+ Group:** RR = 0.11 (95% confidence interval (CI): 0.047 to 0.256).–**Missed Diagnosis Group:** RR = 9.10 (95% CI: 3.90 to 21.23).

These findings emphasize the critical role of DAT-SPECT in facilitating early and accurate diagnosis, which can significantly influence patient management and outcomes. Ba and Martin [18] added that “DAT-SPECT can provide early evidence of presynaptic nigrostriatal pathology, supporting the presence of neurodegenerative presynaptic dopaminergic deficits which is crucial for timely intervention and improved patient outcomes”.

### 3.5. Comparison of Outcomes

The comparison of outcomes between the DAT+ group and the Missed Diagnosis group highlights several key points:**Lower Fall Incidence:** Patients in the DAT+ group had a substantially lower incidence of falls compared to those in the Missed Diagnosis group (Table 1).**Importance of Early Diagnosis:** The data underscores the important role of DAT scans in establishing an earlier diagnosis, which can lead to better management of PD symptoms and a reduction in fall-related injuries.

### 3.6. Multimorbidity Analysis

An upset plot was created to illustrate the extent and complexity of multimorbidities present in the Missed Diagnosis group (Figure 2). This visual representation highlights the diagnostic challenges in this population, where overlapping symptoms from multiple conditions can obscure the clinical diagnosis of PD. A violin plot was generated to compare and contrast the age distribution between the two groups. (Figure 3) showcases the potentially Missed Diagnosis group to encompass a larger range of patient ages in comparison to the DAT+ group. Despite these challenges, prior research supports the high sensitivity and specificity of DAT scans in identifying dopamine deficit conditions [5,17]. This suggests that DAT scans could effectively navigate through the complexity of multimorbidities to more accurately diagnose PD or other dopamine deficiency states, potentially preventing adverse outcomes such as falls.

Key findings from the analysis include the following:**Diagnostic Yield:** DAT scans are effective in confirming dopamine deficiency and hence assisting with the diagnosis of PD in cognitively impaired older patients with a high diagnostic yield. Positive DAT scans are especially helpful when clinical signs and symptoms of PD are mild or vague.**Therapeutic Impact:** Early diagnosis and initiation of dopamine therapy in the DAT+ group resulted in significantly fewer falls compared with the Missed Diagnosis group.**Underdiagnosis and Potential Inadequate Management:** The high incidence of falls in the Missed Diagnosis group indicates the potential underdiagnosis and inadequate or suboptimal management of Parkinsonian symptoms in cognitively impaired patients who did not undergo DAT imaging.

## 4. Discussion

### 4.1. Interpretation of Findings

This study demonstrates the significant role of Dopamine Transporter (DAT) scans in diagnosing Parkinson’s Disease (PD) among cognitively impaired elderly patients. The high diagnostic yield of DAT scans, with 61% of scanned patients showing positive results indicative of PD, highlights their utility in clinical settings where clinical diagnosis alone may be insufficient. This aligns with the findings of Kägi et al. [5], who noted the high sensitivity and specificity of DAT imaging for diagnosing early PD and differentiating it from other non-degenerative Parkinsonian disorders.

The findings highlight the clinical importance of early diagnosis. The significantly lower incidence of falls in the DAT+ group compared to the Missed Diagnosis group (relative risk = 0.11, 95% CI: 0.047 to 0.256) emphasizes the impact of timely intervention. Early detection of PD through DAT imaging and subsequent treatment with dopamine therapy can mitigate adverse outcomes such as falls, which are particularly detrimental in the elderly population. The high relative risk of falls in the Missed Diagnosis group (RR = 9.10, 95% CI: 3.90 to 21.23) indicates that untreated dopamine deficits contribute significantly to the risk of falls.

### 4.2. Clinical Relevance

The findings of this study have important implications for clinical practice. Early diagnosis and prompt initiation of treatment for PD are crucial for managing symptoms and preventing complications. DAT imaging provides a reliable method for identifying dopaminergic deficits early, even in patients with cognitive impairment. This is particularly relevant as cognitive impairment can obscure the clinical diagnosis of PD, leading to underdiagnosis and delayed treatment. PD is frequently misdiagnosed by clinicians when the classic, pill-rolling tremor of PD is absent or the tremor appears like essential tremor. In many cases, the fatigue and general slowness of PD are mistaken for aging. As shown, early diagnosis and treatment in the DAT+ group resulted in a significantly lower incidence of falls compared to in the Missed Diagnosis group. Ba and Martin [18] emphasized that “DAT-SPECT can accurately detect presynaptic dopaminergic deficits, making it useful in the early diagnosis of PD and differentiation from non-degenerative parkinsonian disorders such asessential tremor, dystonic tremor, and drug-induced parkinsonism”. Primary care providers are encouraged to utilize DAT scans more proactively, especially in high-risk populations. Enhanced education and awareness among healthcare providers regarding the benefits of DAT imaging can lead to more timely and accurate diagnoses, ultimately improving patient management and outcomes.

### 4.3. Comparison with the Existing Literature

The results of this study are consistent with previous research indicating the diagnostic and prognostic value of DAT imaging in PD. Ravina et al. [9] demonstrated that lower striatal DAT binding is associated with higher risks of reaching clinical milestones, including motor-related disability, cognitive impairment, and psychosis. Similarly, this study found that the DAT+ group, diagnosed and treated early, had significantly better outcomes in terms of fall incidence compared to the Missed Diagnosis group.

Brooks [20] emphasized the role of DAT imaging in differentiating idiopathic PD from other conditions like vascular parkinsonism. This study’s findings support this view, showing that DAT imaging can effectively identify dopaminergic deficits characteristic of PD even in a population with overlapping cognitive impairments.

### 4.4. Practical Applications in Clinical Practice

Barriers to the practical application of DAT scans in clinical practice have limited their widespread use. These challenges include difficulty in differentiating between Parkinson’s Disease (PD) and atypical Parkinsonian syndromes, the potential for misinterpretation of results—particularly outside specialized centers—and the relatively high costs associated with the scan [21,22]. Additionally, while DAT scans are highly sensitive in detecting dopaminergic deficits, these do not always correlate with disease progression, as evidenced by the phenomenon of ’scans without evidence of dopaminergic deficits’ (SWEDD), where patients exhibit tremors resembling early PD without corresponding DAT scan findings [23]. These factors might diminish the perceived value of DAT scans, especially in settings with strong clinical diagnostic capabilities. Despite these barriers, DAT scans play a crucial role in the early identification of dopaminergic deficits, particularly in cognitively impaired elderly patients who may be at high risk for PD. While not a replacement for thorough clinical assessment, DAT scans can supplement the diagnostic process by highlighting individuals who require closer monitoring or further evaluation. Our study demonstrates that patients who underwent DAT scans had a significantly lower incidence of falls compared to those who did not receive the scan or follow-up interventions, indicative of the prognostic value of early detection in improving patient outcomes. Furthermore, DAT imaging is valuable for preventing the progression of PD-related complications, such as gait and balance issues that contribute to falls. This characterizes DAT scans as a key tool for early intervention, as highlighted by Palermo et al., showing their ability to detect subclinical dopaminergic dysfunctions [19]. Akdemir et al. emphasized the utility of DAT-SPECT imaging in aiding the differential diagnosis of parkinsonism, reinforcing the need for its proactive use in high-risk populations [17]. The integration of DAT imaging into routine clinical practice has the potential to enhance diagnostic accuracy and improve patient outcomes. This benefit is further complemented by using dopaminergic radiotracers in Positron Emission Tomography (PET), such as F-DOPA, which is particularly valuable in differentiating PD from atypical Parkinsonian disorders by allowing for a more detailed assessment of dopamine synthesis and storage capacity in the brain [24]. The combination of DAT-SPECT and F-DOPA PET offers a comprehensive approach to diagnosing and managing Parkinsonian syndromes, thereby facilitating more precise and personalized treatment strategies. We advocate for the use of DAT scans in primary care settings in collaboration with specialists who can interpret the results accurately and guide subsequent evaluation and management plans. From a consumer perspective, concerns about radiation exposure and costs are common barriers to entry. However, in the context of the significant risks associated with falls in older adults—including 38,000 deaths and 3 million emergency department visits in 2021 alone—the benefits of early identification far outweigh these concerns [25]. A longitudinal study conducted for the safety analysis of DAT scans concluded that the procedure was well tolerated, with adverse events being infrequent and mild with the whole-body radiation dose from a DAT scan reaching 3.94 mSv, significantly lower than that of a CT scan, which has 10 mSv of radiation exposure [26]. Most diagnostic modalities in PD are often limited by their over-reliance on the presence of motor symptoms, by which point up to 60 percent of all dopamine neurons within specific regions of the basal ganglia may be lost [27]. Thus, catching the disease at advanced stages limits the implementation of interventions targeted to delay disease progression and offer neuroprotective effects. The goal of disease identification should be to slow down disease progression, treat symptoms where applicable, and preserve the integrity of neuronal connections. The benefits of this are reaped in patient quality of life and cost savings [27]. DAT scans can bridge the gap in the current diagnostic threshold for geriatric cognitively impaired patients presenting for initial assessment for PD in conjunction with non-motor symptoms, which surface much earlier in the disease course. The benefits in quality of life, risk reduction of adverse events linked to symptomatology, and long-term cost savings are contingent upon diagnosing and treating PD before the destructive structural changes have taken place.

### 4.5. Limitations

While this study provides valuable insights, it is not without limitations. The retrospective nature of the analysis may have introduced biases related to data collection and patient selection. Patient demographics were sparsely reported in the dataset, with the exception of age. Additionally, the study did not account for all potential confounding factors that could influence the incidence of falls, such as concurrent medications and comorbidities. Another caveat that needs to be mentioned is that dopamine deficiency does not always confirm Parkinson’s Disease. Dopamine deficiency observed on DAT scans can also be seen in Parkinson’s plus syndromes and could be indicative of multiple system atrophy (MSA), progressive supranuclear palsy (PSP), cortical-basal ganglionic degeneration (CBGD), and dementia with Lewy bodies (DLB). Parkinson’s plus syndromes often do not respond well to dopamine-based or dopamine agonist therapies. Our retrospective dataset lacked information on Parkinson’s plus syndromes, and, even if patients were DAT positive, if they had a Parkinson’s plus syndrome, then they would not have responded well to therapy for Parkinson’s. Although in this study we focused of patients with CI, patients without significant CI might also have features of early PD. The symptomatology of PD can often be confused with other age-related conditions causing fatigue or slowing or aging itself. The upset plot for the Missed Diagnosis group illustrates the extent and complexity of other multimorbidities, emphasizing the diagnostic challenges. Prior research supports the high sensitivity and specificity of DAT scans in identifying dopamine deficiency conditions, suggesting that DAT scans could navigate through the complexity of multimorbidities to more accurately diagnose PD, allow earlier treatment, and prevent adverse outcomes such as falls.

## Figures and Tables

**Figure 1 geriatrics-09-00126-f001:**
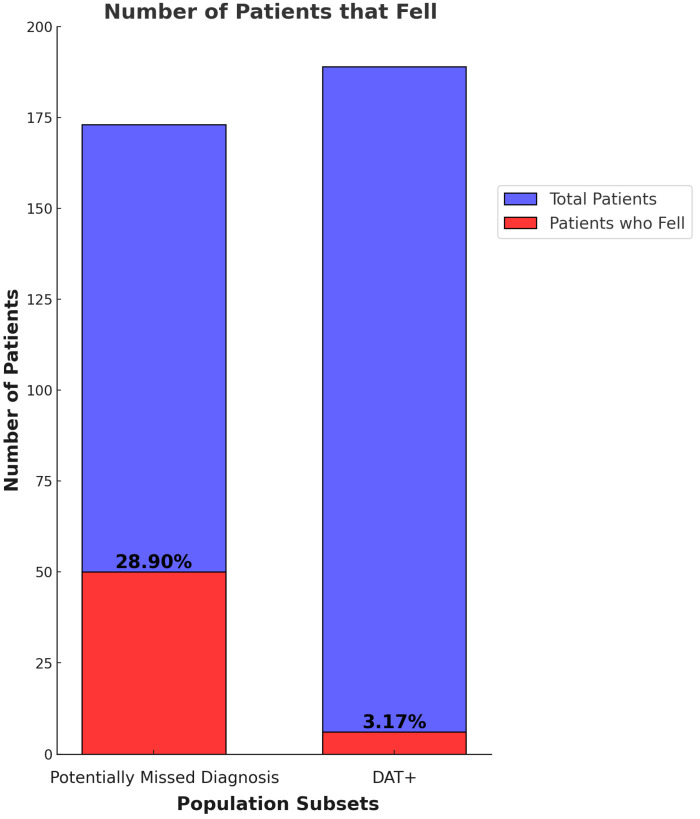
Showcases the difference in the number of patients who experienced falls between the DAT+ and the Missed Diagnosis groups. The significant increase in the percentage of falls could potentially be attributed to unmanaged Parkinsonian symptoms due to a lack of dopamine optimization therapy in the Missed Diagnosis group in comparison to the DAT+ group.

**Figure 2 geriatrics-09-00126-f002:**
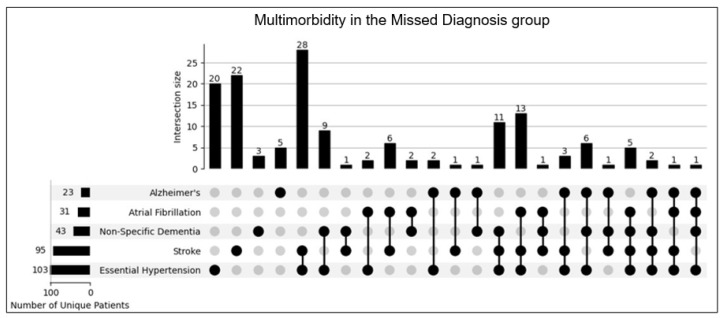
To navigate and view the multimorbidities present in the Missed Diagnosis group, an upset plot was generated to view and quantify the presence of other etiological causes of symptoms.

**Figure 3 geriatrics-09-00126-f003:**
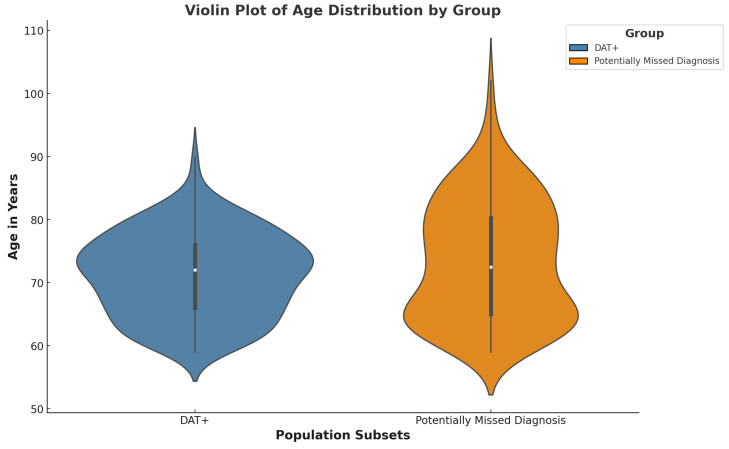
This violin plot illustrates the age distribution of patients in the DAT+ and Missed Diagnosis groups. The width of each violin represents the density of patients at different ages within each group. The plot highlights the spread and central tendency of ages, providing a demographic visual for the two groups.

**Table 1 geriatrics-09-00126-t001:** This table provides a summary of the key statistics comparing the DAT+ group and the Missed Diagnosis group. It includes the total number of patients in the study, the number of patients scanned, positive DAT results, subsequent PD diagnoses, patients with cognitive impairment, dopamine therapy initiation rates, and the incidence of falls. The table highlights that, out of the 297 patients scanned, 189 (64%) had positive DAT results, all of whom were subsequently diagnosed with PD. In the DAT+ group, all 297 patients had cognitive impairment and 89 (47%) received dopamine therapy, resulting in a significantly lower incidence of falls (3%) compared to in the Missed Diagnosis group, where the 173 patients with cognitive impairment experienced a much higher incidence of falls (29%). The relative risk of falls is markedly higher in the Missed Diagnosis group (RR = 9.10, 95% CI: 3.90 to 21.23) compared to the DAT+ group (RR = 0.11, 95% CI: 0.047 to 0.256), emphasizing the critical role of early diagnosis and treatment in managing Parkinsonian symptoms and reducing fall-related injuries.

Category	DAT+ Group	Missed Diagnosis Group
Total Patients in Study	6483	6483
Patients Scanned	297	
Positive DAT Results	189 (64% of scanned)	
Subsequent PD Diagnoses	189 (100% of positive DAT)	
Patients with Cognitive Impairment	297	173
Dopamine Therapy Initiation	89 (47% of positive DAT)	
Falls	6 (3% of DAT+ group)	50 (29% of Missed Diagnosis group)
Relative Risk of Falls	0.11 (95% CI: 0.047 to 0.256)	9.10 (95% CI: 3.90 to 21.23)

## Data Availability

The data presented in this study are available on request from the corresponding author. The data are not publicly available due to privacy restrictions.

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
