# Peer review of "Missed Insights for Earlier Management of Parkinson’s Disease and the Value of Dopamine Transporter (DAT) Scans"

_geriatrics, 2024, doi:10.3390/geriatrics9050126_

Round 1
Reviewer 1 Report
Comments and Suggestions for Authors In some places I would like a more critical presentation.A bit too positive in my opinion. On the one hand, the examination is
expensive and, on the other hand, it involves radiation. That's why you
should look at it a little more critically.
Reviewer 2 Report
Comments and Suggestions for Authors
Hafeez et al performed an analysis on early management of Parkinson’s Disease (PD) and the significance of examination using Dopamine Transporter (DaT) Scans. I have the following comments regarding this manuscript:
1. In the introduction authors bring up the issue concerning PD misdiagnosis, however it would be valuable to acknowledge this aspect in the context of clinical differentiation between PD and atypical parkinsonisms, which is problematic in the early stages - Significance of dysautonomia in Parkinson's Disease and atypical parkinsonisms. Neurol Neurochir Pol. 2024;58(2):147-149. doi: 10.5603/pjnns.98678. Epub 2024 Mar 19. PMID: 38501557, Differentiation of atypical Parkinson syndromes. J Neural Transm (Vienna). 2017 Aug;124(8):997-1004. doi: 10.1007/s00702-017-1700-4. Epub 2017 Feb 27. PMID: 28243754.
2. Authors state: Dopamine deficiency syndromes encompass a range of conditions characterized by reduced dopamine levels in the brain including Parkinson’s Disease (PD), Progressive Supranuclear Palsy (PSP), Multiple System Atrophy (MSA), and Corticobasal Degeneration (CBD). In Corticobasal Syndrome dopamine deficiency may not be necessarily observed due to Alzheimer’s Disease (AD) pathology. This should be acknowledged - Dopamine transporter SPECT imaging in corticobasal syndrome: A peak into the underlying pathology? Acta Neurol Scand. 2022 Jun;145(6):762-769. doi: 10.1111/ane.13614. Epub 2022 Mar 20. PMID: 35307816.
3. Authors emphasize that “the clinical picture of each individual, rather than label them with inaccurate diagnostic categories, such as atypical parkinsonism or PSP mimics – This statement should be additionally stressed in the context of PSP-Parkinsonism Predominant, which is more difficult to differentiate with PD in early stage – recently multiple tools (biochemical, neuroimaging – MRI/SPECT HMPAO) were described in the context of evaluation of this clinical entity
4. Material and methods – authors should indicate which drugs possibly impacting DaTSCAN, which were used by patients
5. Authors indicate - The integration of DAT imaging into routine clinical practice has the potential to enhance diagnostic accuracy and improve patient outcomes.- perhaps this could be discussed also in the context of dopaminergic radiotracers in positron emission tomography e.g. F-DOPA
Reviewer 3 Report
Comments and Suggestions for Authors
This study compares the clinical outcomes in patients with cognitive impairment between two groups; patients with dopamine transporter (DAT) scans and 'without receiving DAT scans or treatment'. The authors report that the former group showed less clinical symptoms, such as falls.
Here are my comments;
1. Line 44-46. This sentence seems locagically strange to me. 'The importance of early detection and timely treatment' should be investigated by clinical follow-up studies, not by the feasibility of diagnostic tools, such as DAT.
2. Line 74-82. The citation 'Moreover, for patients for whom the diagnosis is unclear, clinicians must continue to describe accurately the clinical picture' does not support the authors description 'DAT scans play a vital role'.
3. Line 93-95. This is a reproduce Line 41-43. Either one may not be necessary.
4. Line 112-117. The number of patients is repeated in the Results section.
5. It could be hard to understand 'Missed Diagnosis group', which includes patients who 'did not receive a DAT scan, PD diagnosis, or Dopamine therapy'. For example, 189 out of 297 patients had positive DAT scans. Were the rest of 108 patients included in 'Missed Diagnosis group' due to the lack of PD diagnosis?
6. The Conclusion section almost completely repeats 4.1 and 4.2 of the Discussion section. Either one may be removed.
Comments on the Quality of English Language
Line 65-67. 'highlighting the presence of an issue.' What issue?
Line 87-88. "In clinical practice, DAT SPECT imaging may contribute to the management of patients who do not fully meet the diagnostic criteria of PD or have atypical findings." This citation does not provide any evidence; it exist in the literature itself, so only Akdemir et al. [12] is sufficient.
Line 98-100. Probably a simpler description, such as, 'We retrospectively analyzed the datasets of..'.
Round 2
Reviewer 1 Report
Comments and Suggestions for Authors
thank you
Reviewer 2 Report
Comments and Suggestions for Authors
I do not have further comments.
Reviewer 3 Report
Comments and Suggestions for Authors
The revised version and the authors' responses well address my concerns. I still think 'Missed Diagnosis group' could be simply replaced by 'DAT- group' but the current version would be fine.